# The Effect of Selective Retina Therapy with Automatic Real-Time Feedback-Controlled Dosimetry for Chronic Central Serous Chorioretinopathy: A Randomized, Open-Label, Controlled Clinical Trial

**DOI:** 10.3390/jcm10194295

**Published:** 2021-09-22

**Authors:** Ji-young Lee, Min-hee Kim, Seung-hee Jeon, Seung-hoon Lee, Young-jung Roh

**Affiliations:** 1Department of Ophthalmology and Visual Science, Yeouido St. Mary’s Hospital, College of Medicine, Catholic University of Korea, 10, 63-ro, Yeongdeungpo-gu, Seoul 07345, Korea; jiyounglee.md@gmail.com (J.-y.L.); chriszz@naver.com (M.-h.K.); myname_lsh@hanmail.net (S.-h.L.); 2Department of Ophthalmology and Visual Science, Incheon St. Mary’s Hospital, College of Medicine, Catholic University of Korea, 56, Dongsu-ro, Bupyeong-gu, Incheon 21431, Korea; jsh881107@hanmail.net

**Keywords:** central serous chorioretinopathy, real-time feedback-controlled dosimetry, retinal sensitivity, selective retina therapy

## Abstract

This prospective randomized controlled trial evaluated the safety and efficacy of real-time feedback-controlled dosimetry (RFD)-guided selective retina therapy (SRT) in chronic central serous chorioretinopathy (CSC). Forty-four participants with chronic CSC were included and randomly assigned to the control group or SRT group. The SRT laser system with RFD-guidance was applied to cover the entire leakage area. If SRF remained at the 6-week follow-up visit, re-treatment and rescue SRT was performed for the SRT group and crossover group, respectively. The rate of complete resolution of subretinal fluid (SRF), mean SRF height, and mean retinal sensitivity were compared between the two groups at 6-weeks post-treatment. The complete SRF resolution rate in all SRT-treated eyes was evaluated at 12-weeks post-treatment. The rate of complete SRF resolution was significantly higher in the SRT group (63.6%) than in the control group (23.8%) at 6-weeks post-treatment (*p* = 0.020). The mean SRF height at 6 weeks after SRT was significantly lower in the SRT group (*p* = 0.041). Overall, SRT-treated eyes showed complete SRF resolution in 70.3% of eyes at 12-weeks post-treatment. RFD-guided SRT was safe and effective to remove SRF in chronic CSC patients during the 3-month follow-up period.

## 1. Introduction

Central serous chorioretinopathy (CSC) is a chorioretinal disease associated with serous retinal detachment caused by choroidal hyperpermeability and one or more areas of leakage through a retinal pigment epithelium (RPE) defect [1,2]. Although debated [3,4], many studies have classified the condition into acute and chronic CSC, based on the duration of serous retinal detachment and irreversible tissue changes, such as RPE atrophy [4,5]. Most patients with acute CSC show spontaneous resolution of serous retinal detachment within 3 months [6], but chronic CSC patients have persistent subretinal fluid (SRF), which may cause irreversible tissue damage and long-term visual sequelae [2,3,7].

CSC treatment aims to eliminate the underlying SRF and achieve complete resolution of serous retinal detachment [8]. There is currently no consensus on CSC treatment guidelines, given its poorly understood pathophysiology and lack of a unified CSC classification system [1,3]. Several treatment modalities, including conventional laser photocoagulation, photodynamic therapy (PDT), and intravitreal anti-vascular endothelial growth factor (anti-VEGF) have been used clinically, but with several adverse events. The application of a conventional laser was limited to extrafoveal leakage areas because of complications, such as central scotoma, loss of contrast sensitivity, and choroidal neovascularization development [2,9]. Although PDT using verteporfin showed good CSC resolution, RPE atrophy and choroidal ischemia have been reported [10,11]. The intravitreal anti-VEGF injection has anti-proliferative and anti-hyperpermeability effects under experimental conditions; however, injection of anti-VEGF is an off-label use for CSC, and its effects on CSC remain controversial [12,13,14].

To avoid retinal tissue damage due to conventional laser photocoagulation, subthreshold laser modalities implementing “micropulses” rather than continuous waves have been used for treating CSC [15]. Selective retina therapy (SRT), a micropulse laser modality, has been investigated for treating CSC. Unlike other subthreshold micropulse lasers (SMPLs), which do not cause any retinal tissue damage, SRT induces selective RPE damage without affecting the overlying photoreceptors [15]. Since the microsecond (1.7-μs) SRT pulses are primarily absorbed by melanosomes inside RPE cells, SRT induces the formation of microbubbles around the melanosomes. Microbubbles cause RPE damage and trigger RPE cell migration and proliferation. Restoration of the RPE layer by proliferating RPE cells is a crucial process in RPE rejuvenation [16], which may involve various cell mediators [17,18].

In previous studies, adequate SRT-spot energy was determined by two therapeutic endpoints including a “visible spot on fundus fluorescein angiography (FFA)” and an “invisible spot on color fundus photography (CFP)” [19,20,21,22,23]. To determine the appropriate pulse energy without using FFA for SRT, real-time feedback-controlled dosimetry (RFD) has been developed and optimized for various macular diseases, including CSC [21,22,23], diabetic macular edema [24], and intermediate age-related macular degeneration (AMD) [25]. RFD with both optoacoustic dosimetry and reflectometry detects transient microbubbles originating from RPE damage in real time and allows individualized laser settings, ensuring the safety of SRT. Optoacoustic dosimetry detects ultrasonic pressure-wave signals from microbubbles, producing an optoacoustic feedback value [26,27], and reflectometry detects the modulation of backscattered light signals from the microbubbles, producing an optical feedback value [28,29,30].

Although RFD-guided SRT for CSC patients has shown favorable results in previous studies [21,22], pretreatment FFA was used simultaneously for test spots to achieve adequate preset pulse energy. Since the rate of detection of RPE damage by RFD for CSC and intermediate AMD was 95.7% and 93.5%, respectively, SRT using RFD sometimes caused undertreatment rather than overtreatment, such as a burn, as previously reported [22,25]. In this prospective randomized controlled trial (RCT), we evaluated the safety and efficacy of RFD-guided SRT, without pretreatment FFA, in chronic CSC. Based on our experience with RFD, pretreatment FFA was not used to determine the preset pulse energy in the present study, while RFD was used only for dosing the pulse energy.

## 2. Methods

This prospective, open-label RCT was conducted at a single center in Seoul, South Korea. This study was approved by the Institutional Review Board of the Catholic University of Korea (SC17DESV0089, Seoul, South Korea) and conducted in accordance with the tenets of the Declaration of Helsinki. This study was registered at the Clinical Research Information Service in South Korea. (KCT0006325) All participants provided written informed consent after having been informed about the possible risks of SRT.

### 2.1. Participant Enrollment

We enrolled participants with chronic CSC who had (1) eyes with persistent (≥3 months) subretinal fluid (SRF) involving the fovea on optical coherence tomography (OCT) images; (2) focal or diffuse active hyperfluorescent leakages due to CSC on fundus fluorescein angiography (FFA) images; and (3) were aged 19–65 years. Exclusion criteria were as follows: (1) Eyes with best-corrected visual acuity (BCVA) of 20/200 or worse (Snellen equivalent); (2) other chorioretinal diseases, such as choroidal neovascularization and polypoidal choroidal vasculopathy; (3) lens or vitreous opacity hampering retinal imaging and laser treatment; (4) RPE atrophy exceeding 1000 µm in diameter, involving the fovea; (5) large retinal pigment epithelial detachment (PED) exceeding 300 µm in width or 100 µm in height; (6) history of conventional laser photocoagulation or PDT for CSC; (7) history of intravitreal injection of anti-VEGF within 10 weeks pre-SRT; (8) use of systemic, periocular, or intraocular corticosteroids within 1 year pre-SRT; (9) use of spironolactone, acetazolamide, or ketoconazole within 2 months pre-SRT. The participants were randomly assigned, in a 1:1 ratio, to the control or SRT group.

### 2.2. SRT Procedure

A single retinal specialist (YJR) performed all treatments using the SRT laser system with RFD-guidance (R:GEN, Lutronic, Goyang-si, South Korea). An Nd:YLF laser with a 527-nm wavelength and a 200-µm spot diameter was used to deliver 15 micropulses of 1.7-µs per spot. The pulse repetition frequency was 100 Hz. The RFD was equipped with a photodiode as a reflectometric sensor and a contact lens (field-of-view, 90 D; image magnification × 1.05) with an ultrasonic transducer inserted as an acoustic sensor. The SRT device received a CE mark from the EU Notified Body and was approved for CSC by the Ministry of Food and Drug Safety of South Korea.

Each individual SRT spot involved a burst of, at most, 15 micropulses of laser irradiation, ramped linearly by an increment of 3.57% for the following micropulse. The first micropulse was programmed to be 50% of the energy of the last (15th) micropulse. The physician could determine the preset pulse energy by adjusting the 15th micropulse, as previously described [22,24]. The RFD had fixed thresholds, comprising 2.0 arbitrary units (AU) for the optoacoustic sensor and 6.0 AU for the reflectometry sensor [21,22,24,25]. The RFD algorithm was programmed to avoid under- and over-treatment: It automatically ceased irradiation immediately for each individual spot when either an optoacoustic value > 2.0 AU, or a reflectometry value > 6.0 AU, was obtained, which we named auto-stop. The preset pulse energy for SRT spots was adjusted by the surgeon based on the on-screen arrow signals of the RFD during each irradiation. Briefly, if an upward-pointing arrow, indicating undertreatment, was shown, the preset pulse energy was increased in 10-µJ steps. If a downward-pointing arrow, indicating overtreatment, was shown, the preset pulse energy was decreased in 10-µJ steps, as previously described [22,23,24,25]. The surgeon controlled the preset pulse energy until a sideways-pointing arrow, indicating adequate pulse energy, was presented on the screen. To determine the margin of safety of the preset pulse energy for each patient, 10–15 preliminary test spots with escalating pulse energy (80–200 µJ) were applied at the adjacent area of the superior or inferior temporal arcade vessels. Among the preset pulse energies indicating sideways-pointing arrows, the minimum pulse energy was initially chosen for the treatment spot irradiation. If test spots were visible during the test spot irradiation, the highest level of the preset pulse energy for the treatment spot was set to be lower than the pulse energy of the visible spots. The optoacoustic and reflectometry values of treatment spots in the SRT group were analyzed to evaluate which dosimetry caused the auto-stop. Although the SRT spots were invisible during irradiation, a one-spot spacing-density was maintained using a guide beam.

SRT spots were applied to cover the entire leakage area on FFA. If SRF remained at 6-weeks post-treatment in the SRT group, retreatment was performed with the same initial preset pulse energy as in the first SRT. If SRF remained at the 6-weeks follow-up in the control group, rescue SRT was performed for the crossover group.

### 2.3. Clinical Outcome Evaluations

Each participant underwent complete ophthalmological examinations, including best-corrected visual acuity (BCVA), slit-lamp examination after pupil dilation, CFP (CF-60UVi, Canon Inc., Ōta, Japan), fundus autofluorescence (FAF) (HRA2, Heidelberg Engineering, Dossenheim, Germany), swept-source OCT (DRI OCT Triton, Topcon, Tokyo, Japan), FFA, and indocyanine green angiography at baseline. SRF height (maximum distance between the outer neurosensory retina and the RPE), central macular thickness (CMT), and central choroidal thickness (CCT) were measured using the OCT device. Scanning was performed using a 7 × 7-mm^2^ volume centered on the fovea, and retinal layers were identified using IMAGENET 6.0, software (Topcon). FFA and ICG images were reviewed and categorized into focal (1–3 leakage points) and diffuse (≥4 leakage points) types according to the number of leakage points. FAF images were evaluated and classified into hyperautofluorescent or hypoautoflurescent by comparing them with the background intensity at serous retinal detachment lesions. The classification was determined based on the average value of a histogram of serous retinal detachment lesions using an image analysis program (Adobe Photoshop 7.0, Adobe Systems Inc., San Jose, CA, USA) as previously described [31].

To evaluate the effect of SRT, all participants underwent logMAR BCVA, slit-lamp examination, CFP, and OCT examinations at 3-, 6-, and 12-weeks post-treatment. Additionally, retinal sensitivity was measured using microperimetry (MAIA-TM; Macular Integrity Assessment, CenterVue SpA, Padova, Italy) at baseline and at 6 weeks, under the following settings: A fixation target of a 2°-diameter red ring, a dynamic stimulus range of 0–36 dB, Goldmann III stimulus size, white and monochromatic background at 4 apostilb, 200-ms stimulus duration, and 1000-apostilb maximum luminance. The standard 10° grid with 37 stimuli consisted of retinal sensitivity of a central point, and three concentric rings representing 2°, 6°, and 10°, with 12 test points each (4-2 threshold strategy). The first and second microperimetry tests were performed on the screening day, and data from the second test were used for statistical analysis.

The primary outcome measure was the rate of complete resolution of SRF on OCT between the two groups at 6-weeks post-treatment. The secondary outcome measures, including the change in SRT height, CMT, CCT, logMAR BCVA, and retinal sensitivity on microperimetry, were assessed 6 weeks after the initial treatment. Additionally, at 12-weeks post-treatment, we evaluated the complete SRF resolution rate of SRT-treated eyes, including in the SRT and crossover groups.

### 2.4. Statistical Analysis

For sample size calculation, we considered that the SRT treatment group would be superior to the control group in terms of the percentage of participants who showed complete SRF resolution after 6 weeks. The level of significance was 5% with a one-sided test, power was 80%, and the superiority margin was set to zero according to the European Medicines Agency guidelines. The rate of SRF resolution was measured in the range of 0.71–1.0 in the SRT and at 0.4 in the control group [19,20,21,22]. Calculations showed that it was necessary to register at least 20 subjects in each group, assuming a dropout rate of 10%. Thus, 44 subjects were enrolled.

To compare baseline characteristics and clinical outcomes between the two groups, the *t*-test was used for continuous variables and the chi-square test for categorical variables. The differences in measured variables (SRF height, CMT, CCT, logMAR BCVA, and retinal sensitivity on microperimetry) between baseline and a specific time-point were evaluated using a paired *t*-test. Repeated-measures analysis of variance with post-hoc analysis with Bonferroni correction was used for repeated analyses of the same variables. Statistical analysis was performed using SPSS (version 22.0; SPSS Inc., Chicago, IL, USA). Statistical significance was set at *p* < 0.05.

## 3. Results

Of the 58 patients assessed for screening, 14 were excluded and 44 were included based on the inclusion and exclusion criteria in the study and were randomly and equally assigned to the SRT-treated (SRT group) and the control group. One control participant dropped out after the first visit (at 3 weeks) and another control participant dropped out after the second visit (at 6 weeks). Thus, 22 and 20 participants in the SRT and control groups, respectively, completed the study (Figure 1). There were no significant differences between the control and SRT groups in terms of baseline characteristics (Table 1).

The rate of complete SRF resolution was significantly higher in the SRT group (63.6%) than in the control group (23.8%) at 6-weeks post-treatment (*p* = 0.020) (Figure 2). The mean SRF height at 6 weeks after SRT was significantly lower in the SRT than in the control group (*p* = 0.041). However, the mean CMT, mean CCT, mean BCVA, and mean retinal sensitivity were similar in the two groups at 6-weeks post-treatment (Table 2).

While mean SRF height (173.4 ± 88.6 μm) at baseline decreased significantly to 78.5 ± 89.8 μm at 3-weeks and 79.2 ± 130.4 μm at 6-weeks post-treatment in the SRT group (all *p* < 0.001), it did not change significantly in the control group (*p* = 0.137). In the SRT group, the mean CMT reduced significantly from baseline to 3 weeks and 6 weeks (*p* < 0.001, *p* = 0.004, respectively), but not in the control group (*p* = 0.243) (Figure 3, Table 3). The change in the mean CCT of the SRT and control groups was not significant at 6-weeks post-SRT (*p* = 0.324, *p* = 0.506). The mean logMAR BCVA of the SRT and control groups did not improve significantly during the 6-week follow-up (*p* = 0.549, *p* = 0.186) (Table 3). While there was a significant increase in the mean retinal sensitivity from 19.5 ± 3.7 dB at baseline to 22.3 ± 4.0 dB at 6-weeks post-treatment in the SRT group (*p* = 0.002), the change in the mean retinal sensitivity from baseline to 6 weeks was not significant in the control group (*p* = 0.234). Additionally, no scotomatous changes (>6 dB decrease) were observed at any test point within the central 10° during the 6-week follow-up period.

Since 76.2% (16/21) of eyes showed remaining SRF in the control group at 6 weeks, 16 eyes in the crossover group underwent rescue SRT (Figure 4). In the SRT group, eight eyes with remaining SRF underwent a second SRT at 6-weeks post-SRT (Figure 5). In the crossover group, 10/15 eyes showed complete SRF resolution at 6 weeks after the rescue treatment (Figure 6). In the SRT group, 3/8 eyes showed complete SRF resolution at 6 weeks after the second SRT. Consequently, 70.3% (26/37) of SRT-treated eyes, including the SRT and crossover groups, showed complete SRF resolution at the 12-week follow-up visit. In the crossover group, mean SRF height and CMT decreased significantly after rescue SRT (*p* < 0.001, *p* = 0.002, respectively). The mean logMAR BCVA improved significantly at post-rescue SRT 6 weeks (*p* = 0.028). In the retreatment group, mean SRF height decreased from 217.9 ± 128.3 μm to 138.1 ± 48.8 μm 6 weeks after the second SRT (Table 4).

The mean number of initial SRT spots in the SRT group and rescue treatment spots in the crossover group was 17.2 ± 12.1 and 19.6 ± 13.9, respectively. The mean preset pulse of treatment spots and rescue treatment spots was 135.5 ± 22.2 μJ and 139.7 ± 21.7 μJ, respectively (*p* = 0.553) (range: 115–180 μJ). The mean actually applied pulse energy of treatment spots and rescue treatment spots was 95.5 ± 28.2 μJ and 97.1 ± 30.5 μJ, respectively (*p* = 0.475). All SRT spot lesions were invisible on CFP during the 12-week follow-up period. Among 379 treatment spots of the SRT group, 10 spots were excluded because of defocusing errors. Evaluation of optoacoustic and reflectometry signals of 369 spots showed that auto-stops occurred in 222 spots (60.1%) by optoacoustic signal, in 9 spots by reflectometry (2.4%), and in 102 spots (27.6%) by both signals. Thirty-six of 369 spots (9.8%) that were below the thresholds of both signals had no auto-stops (Figure 7).

Although conjunctival hyperemia was temporarily observed in four participants in the SRT group, no participant showed serious SRT-related adverse events, such as retinal hemorrhage or retinal burn-like discoloration, during the 12-week follow-up period.

## 4. Discussion

In chronic CSC, persistent SRF is associated with damage to the RPE and the overlying photoreceptors, causing irreversible vision loss [1,3]. Although numerous interventions have been introduced to stop this progressive damage and to remove SRF, the optimal treatment option for chronic CSC remains unclear [32]. SRT has been reported as an alternative treatment option for patients with acute and chronic CSC, with favorable anatomical and functional outcomes [19,20,21,22,23]. In this prospective RCT of the safety and efficacy of RFD-guided SRT in chronic CSC, we did not perform pretreatment FFA, and RFD was used only for dosing the pulse energy. As a primary endpoint in the current study, the complete SRF resolution rate was significantly higher in the SRT (63.6%) than in the control group (23.8%) (*p* = 0.020) 6-weeks post-SRT, indicating the efficacy of RFD-guided SRT.

In previous reports, the SRF resolution rate of chronic CSC with ≥ 3 months SRF was in the range of 65–75% [19,21,22]. While SRT without RFD was used for CSC in a previous RCT, 71.4% (10/14) of SRT-treated eyes showed complete SRF resolution, as compared to 40% (6/16) of untreated control eyes at 3-months post-SRT [19]. Although most previous studies measured clinical outcomes at 3-months post-SRT, our results at 3- and 6-weeks post-SRT indicated that a higher rate of complete SRF resolution, at an earlier time point, was achieved in the SRT group than in the control group. In both groups, the reduction in SRF height was significant at 3-weeks post-SRT; however, at 6-weeks post-treatment, a significant SRF reduction was observed only in the SRT group. Since the SRF height of the control group was increased at 6 as compared to 3 weeks, the spontaneous SRF resolution of the control group did not continue until 6 weeks.

In previous reports, acute CSC usually presented with one or a few focal leakages, which resolved spontaneously in 84% of patients at the 6-months follow-up [33]. However, diffuse leakages, as one of the clinical features of chronic CSC, tend to have a lower spontaneous resolution rate. A previous randomized study of SRT showed that control patients, with a rate of 37.5% of diffuse leakages, showed a lower spontaneous SRF resolution rate (40%) by 3 months [19]. In the current study, although the SRF resolution rate was measured earlier than in previous studies, the control group, which involved 63.6% diffuse leakages, showed a relatively lower spontaneous SRF resolution rate (23.8%) by 6 weeks. We suspect that a higher proportion of diffuse leakages might negatively influence spontaneous SRF resolution in this study. Additionally, since 66.7% of the crossover group showed complete SRF resolution at 6 weeks after rescue SRT, the SRF resolution rate in SRT-treated eyes increased to 70.3% by 12 weeks. As SRF recurrence was observed in one patient in the SRT group at 12-weeks post-treatment, 73% of patients experienced complete SRT resolution by 12 weeks. Consequently, RFD-guided SRT showed a similar complete SRF resolution rate as a previous report that did not use RFD [19]. Moreover, at 3-weeks and 6-weeks post-treatment, significant reductions in CMT and SRF height were observed in the SRT group, indicating that RFD-guided SRT was fast and effective in this study.

For treating CSC, the efficacy of other laser modalities, such as SMPL and PDT, has been reported [15,32]. In the PLACE trial, the half-dose PDT group showed a higher complete SRF resolution rate (67%) than the SMPL group (29%) at 6–7 months after treatment [34]. SRT could be considered as a type of SMPL, because the lasers have common characteristics, such as “invisible laser on CFP” and “photoreceptor-sparing micropulse laser”; however, SRT uses much shorter micropulses and different wavelengths. Moreover, SMPL does not produce any retinal damage, whereas SRT induces selective RPE damage. Since the mechanism of SRT is known to be associated with restoration of new RPE rather than the “reset theory” of SMPL [15,35], our results cannot be applied to the effect of other SMPLs.

As previous reports showed no decrease in retinal sensitivity after SRT [21,36], in the current study, the absence of scotomatous change at test points represents the absence of photoreceptor damage in the SRT-treated area. Furthermore, the significantly improved retinal sensitivity at 6-weeks post-treatment might be associated with fast, complete SRF resolution in the SRT group. Additionally, although Kyo et al. reported that the CCT decrease was significant 6 months after SRT [37], in the current study, the CCT of neither the SRT nor the control groups changed significantly during the 6-week follow-up period. As a previous report also showed no significant change in CCT after SRT [22], SRT may primarily affect the RPE layer, rather than the choroidal layer.

While RFD was used to titrate the preset pulse energy during irradiation without pretreatment FFA in the current study, RFD-guided irradiation did not produce any visible changes in all the SRT spots. Although auto-stops occurred according to the signals from both dosimetries simultaneously in 27.6% of the spots, most of the other spots had auto-stops related to the optoacoustic signal (60.1%) rather than the reflectometry signal (2.4%). Therefore, the auto-stop of RFD was mainly performed by optoacoustic dosimetry, rather than reflectometry. As in previous reports on RFD [21,22,25], the mean actually applied pulse energy of treatment spots and rescue treatment spots was lower than the mean preset pulse energy, due to the RFD auto-stop. In addition, RFD-guided SRT resulted in no visible test spots, and therefore no visible treatment spots, because the overtreatment alarm (downward-pointing arrow) of RFD warned the physician to stop increasing the pulse energy of spots. The RFD-guided SRT was effective in avoiding overtreatment and no visible spots were observed during the 12-week follow-up period.

Our study had several limitations. First, the number of patients was relatively small. However, considering the low incidence of CSC, the superiority of the SRT group over the control group was clearly presented in 44 patients. Second, a single-center trial might be associated with biases such as overestimation [38] and surgeon’s skill compared with a multicenter trial. Third, several known risk factors causing CSC, including smoking habits, alcohol consumption, and personality traits, were not investigated. Fourth, the follow-up period was short. Since spontaneous SRF resolution is possible during a long-term follow-up period, we tried to minimize the bias caused by the effect of spontaneous SRF resolution by comparing the early response to SRT between the two groups. Fifth, the accuracy of RFD was not calculated because post-SRT FFA was not performed in this study. Although the SRT test spots can be clearly detected by FFA in CSC, calculating the RFD accuracy through treatment spots is challenging since SRF at the macular area blurs the leaks from SRT spots when there are overlaps with leaks from spots. As RFD accuracy exceeded 90% based on evaluating test spots in previous reports [22,25], it can be considered that RFD-guided SRT was effective for removing SRF without performing FFA. Since repeated FFA examination could be burdensome for patients because of several side effects, RFD-guided SRT could be useful to waive the need for FFA when treating CSC. However, preliminary test spot evaluation by physicians remains mandatory to find the range of appropriate pulse energy in advance of RFD-guided irradiation.

## 5. Conclusions

In conclusion, our results demonstrated that SRT with RFD was safe and effective in resolving SRF over the short-term period. However, further larger RCTs with long-term follow-up are necessary to investigate the effect of SRT with RFD and the recurrence rate in chronic CSC.

## Figures and Tables

**Figure 1 jcm-10-04295-f001:**
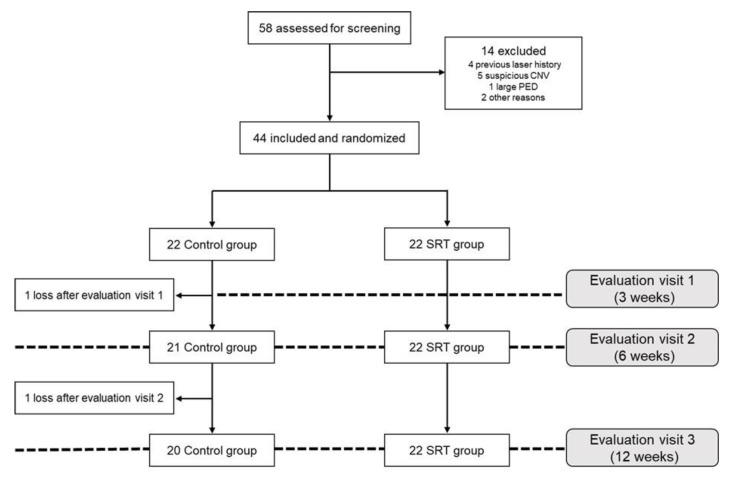
Flow-diagram showing patient allocation to the SRT and control group.

**Figure 2 jcm-10-04295-f002:**
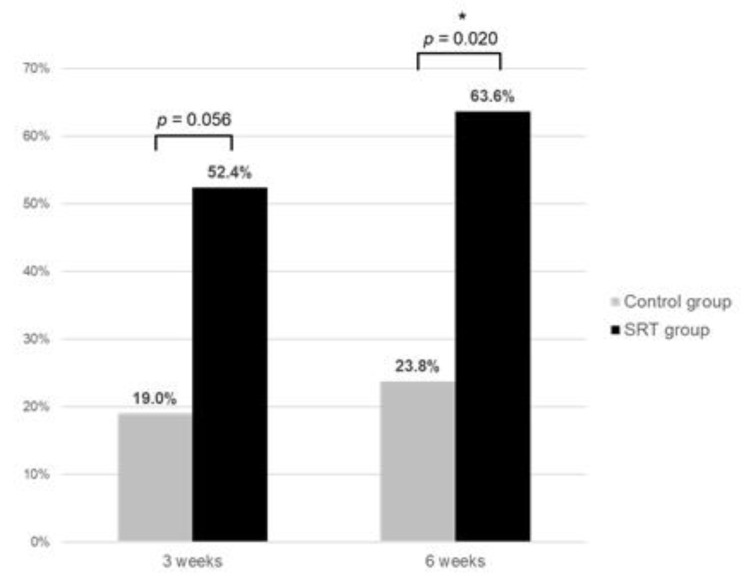
Comparison of the rate of complete resolution of subretinal fluid between the SRT group and control group at 3-weeks and 6-weeks posttreatment. * *p* < 0.05.

**Figure 3 jcm-10-04295-f003:**
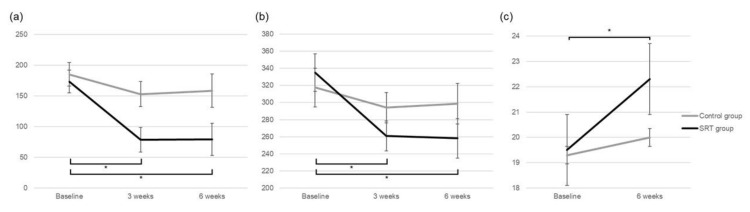
Comparison of change in (**a**) subretinal fluid height, (**b**) central macular thickness, and (**c**) mean retinal sensitivity between the SRT group and control group during the 6-week follow-up period. * *p* < 0.05.

**Figure 4 jcm-10-04295-f004:**
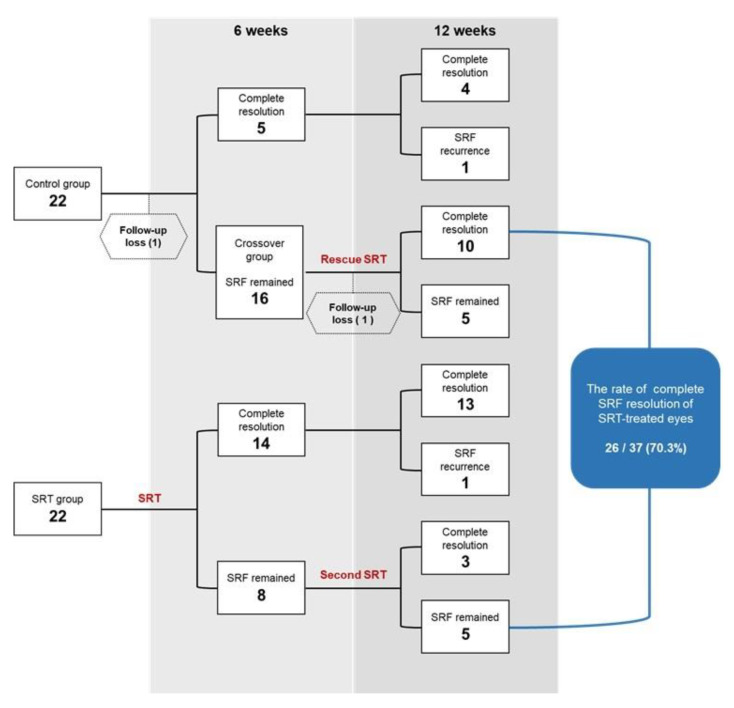
Flow-chart representing the rate of complete resolution of subretinal fluid (SRF) in the SRT group and control group during the 12-week follow-up period.

**Figure 5 jcm-10-04295-f005:**
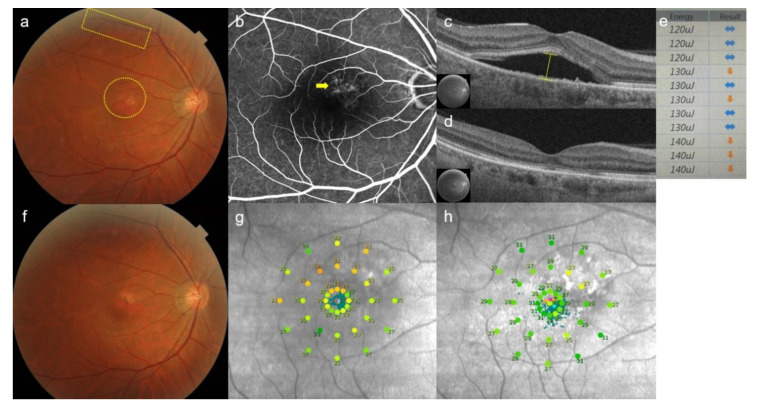
Case 1: A 51-year-old man presented with a 9-month history of blurred vision in the left eye. Subretinal fluid at baseline was observed at the macula, while 11 test spots (yellow rectangular) and 17 treatment spots (yellow circle) were invisible on color fundus photography (**a**). Diffuse leakages (yellow arrow) were shown on fundus fluorescent angiography (**b**). Subretinal fluid height (yellow line) was measured by using optical coherence tomography (OCT) (**c**). At 6-weeks post-selective retina therapy (SRT), subretinal fluid was completely resolved on OCT (**d**). Screen image of real-time feedback-controlled dosimetry (RFD) after test spot irradiation. Based on the algorithm of RFD, the sideways-pointing arrow indicates appropriate pulse energy. The downward-pointing arrow indicates an alarm for overtreatment. The pulse energy of 130 µJ yields different signals (sideways-pointing and downward-pointing arrows), because auto-stops occurred at different places of micropulses by RFD for each spot. The initial pulse energy for a treatment spot was 120 µJ, as the pulse energy constantly showed appropriateness (sideways-pointing arrow). (**e**). No SRT spots were visible on color fundus photography (**f**). Mean retinal sensitivity (17 dB) on microperimetry at baseline (**g**) was improved to (25 dB) on microperimetry (**h**) at 6 weeks after treatment.

**Figure 6 jcm-10-04295-f006:**
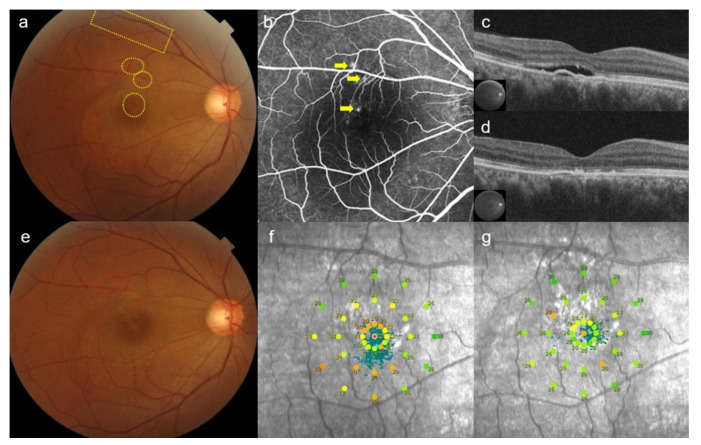
Case 2: A 45-year-old man presented with an 8-month history of blurred vision in the right eye. After the 6-week observation period, subretinal fluid was still observed at the macula on color fundus photography. The area of 13 test spots (yellow rectangular) and 15 rescue treatment spots (yellow circles) were invisible (**a**). Three focal leakages (yellow arrows) were shown on fundus fluorescent angiography (**b**). Subretinal fluid and pigment epithelial detachment was observed on optical coherence tomography (OCT) before rescue with selective retina therapy (SRT) (**c**). At 6-weeks post-SRT, subretinal fluid was completely resolved on OCT (**d**). No SRT spots were visible on fundus photography (**e**). Mean retinal sensitivity (20 dB) on microperimetry at baseline (**f**) was increased to (22 dB) on microperimetry (**g**) at 6-weeks post-treatment.

**Figure 7 jcm-10-04295-f007:**
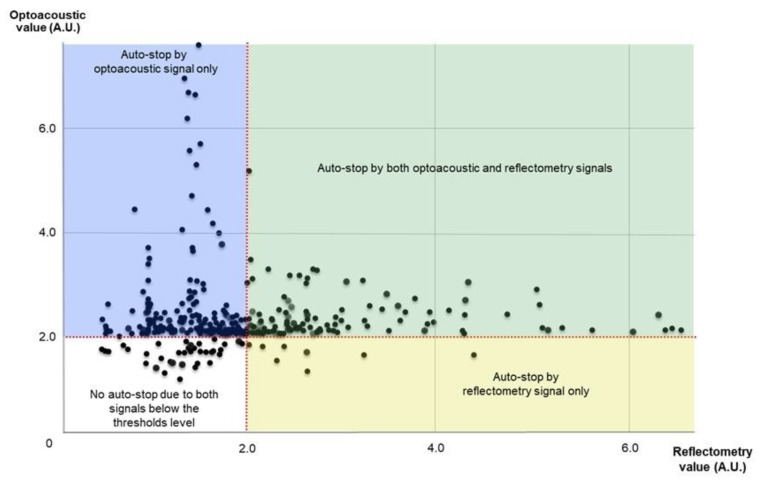
Scatter plot of optoacoustic and reflectometry values in selective retina therapy treatment spots.

**Table 1 jcm-10-04295-t001:** Baseline demographics of participants.

	Control Group (*n* = 22)	SRT Group (*n* = 22)	*p*-Value
Age (years)	47.1 ± 9.1	43.6 ± 7.7	0.181
Sex, male/female (*n*)	19/3	17/5	0.696
Laterality, OD/OS (*n*)	14/8	13/9	1.000
Type of leakages by FFA (*n*, focal: diffuse)	8:14	6:16	1.000
FAF pattern at SRD (*n*, hyper-/hypo-autofluorescent)	13/9	14/8	0.757
Duration of symptom (months)	12.9 ± 8.1	13.7 ± 12.7	0.822
Prior intravitreal anti-VEGF injection, *n* (%)	10 (45.5%)	9 (40.9%)	1.000
BCVA (LogMAR)	0.20 ± 0.14	0.25 ± 0.22	0.428
Central macular thickness (μm)	316.1 ± 94.0	335.0 ± 110.5	0.545
Subretinal fluid height (μm)	186.5 ± 84.1	173.4 ± 88.6	0.617
Central choroidal thickness (μm)	369.8 ± 76.2	351.4 ± 63.0	0.566
Mean retinal sensitivity (dB)	19.3 ± 3.8	19.5 ± 3.7	0.904

VEGF, Vascular endothelial growth factor; BCVA, best-corrected visual acuity; FFA, fundus fluorescein angiography; FAF, fundus autofluorescence; SRD, serous retinal detachment.

**Table 2 jcm-10-04295-t002:** The comparison of clinical outcomes between the control and selective retina therapy (SRT)-treated group 6 weeks after SRT.

	Control Group (*n* = 21)	SRT Group (*n* = 22)	*p*-Value
Mean SRF height (μm)	158.5 ± 115.8	79.2 ± 130.4	0.041 *
Mean CMT (μm)	298.6 ± 91.9	258.2 ± 121.9	0.228
Central choroidal thickness (μm)	373.4 ± 71.8	348.1 ± 58.4	0.211
BCVA (LogMAR)	0.18 ± 0.14	0.23 ± 0.23	0.344
Mean retinal sensitivity (DB)	20.0 ± 4.2	22.3 ± 4.0	0.085
Complete resolution rate of SRF (%)	5/21 (23.8%)	14/22 (63.6%)	0.009 *

* *p* < 0.05. SRF, subretinal fluid; CMT, central macular thickness.

**Table 3 jcm-10-04295-t003:** Change in subretinal fluid height, central macular thickness, and best-corrected visual acuity at baseline, 3-weeks post-treatment, and 6 weeks post-treatment in the selective retina therapy (SRT) group and control group.

	SRF Height (μm)		CMT (μm)		CCT (μm)		BCVA (LogMAR)	
	Control Group	SRT Group	Control Group	SRT Group	Control Group	SRT Group	Control Group	SRT Group
Baseline	186.5 ± 84.1	173.4 ± 88.6	316.1 ± 94.0	335.0 ± 110.5	369.8 ± 76.2	351.4 ± 62.9	0.20 ± 0.14	0.25 ± 0.22
3 weeks (*p*-value)	153.0 ± 95.2 (0.036 *)	78.5 ± 89.8 (<0.001 *)	293.2 ± 79.5 (0.109)	261.0 ± 81.5 (<0.001 *)	365.1 ± 70.4 (0.298)	354.3 ± 59.9 (0.361)	0.20 ± 0.14 (0.266)	0.25 ± 0.24 (1.000)
6 weeks (*p*-value)	158.5 ± 115.8 (0.137)	79.2 ± 130.4 (<0.001 *)	298.6 ± 91.9 (0.243)	258.2 ± 121.9 (0.004 *)	373.4 ± 71.8 (0.506)	348.1 ± 58.4 (0.324)	0.18 ± 0.14 (0.186)	0.23 ± 0.23 (0.549)

* *p* < 0.05. SRF, subretinal fluid; CMT, central macular thickness; CCT, central choroidal thickness; BCVA, best-corrected visual acuity.

**Table 4 jcm-10-04295-t004:** Change in subretinal fluid height, central macular thickness, and best-corrected visual acuity in the crossover group (*n* = 15 eyes) and retreatment group (*n* = 8 eyes).

	Mean SRF Height (μm)		Mean CMT (μm)		Mean CCT (μm)		Mean BCVA (LogMAR)	
	Crossover Group	Retreatment Group	Crossover Group	Retreatment Group	Crossover Group	Retreament Group	Crossover Group	Retreatment Group
6 weeks	213.4 ± 83.1	217.9 ± 128.3	330.9 ± 85.7	369.6 ± 145.1	373.6 ± 83.9	344.1 ± 53.5	0.20 ± 0.15	0.41 ± 0.26
12 weeks (*p*-value)	52.7 ± 82.9 (<0.001 *)	138.1 ± 48.8 (0.153)	225.8 ± 62.1 (0.002 *)	333.8 ± 161.9 (0.474)	371.0 ± 83.8 (0.535)	347.1 ± 43.5 (0.730)	0.16 ± 0.14 (0.028 *)	0.40 ± 0.28 (0.351)

* *p* < 0.05. SRF, subretinal fluid; CMT, central macular thickness; CCT, central choroidal thickness; BCVA, best-corrected visual acuity.

## Data Availability

Data available on request due to restrictions e.g., privacy or ethical.

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
