# Peer review of "The Effect of Selective Retina Therapy with Automatic Real-Time Feedback-Controlled Dosimetry for Chronic Central Serous Chorioretinopathy: A Randomized, Open-Label, Controlled Clinical Trial"

_jcm, 2021, doi:10.3390/jcm10194295_

Round 1
Reviewer 1 Report
To confirm the correct dosing of the energy made with RFD, it would be interesting if you could evaluate, at the clinical controls, the presence of any visible spot sign on a autofluorescence examination
Author Response
Reviewer 1
To confirm the correct dosing of the energy made with RFD, it would be interesting if you could evaluate, at the clinical controls, the presence of any visible spot sign on a autofluorescence examination
Response) Thank you for reviewer’ comments. Since the change of SRT spot on autofluorescence examination is absent right after irradiation, the dosing of the pulse energy based on the change of autofluorescence cannot be used for SRT treatment. The change of SRT spots on autofluorescence usually occurs several weeks after SRT treatment. However, for the future study, evaluation for the long-term change of SRT spots on autofluorescence might be helpful to investigate the dosing for adequate pulse energy in some clinical viewpoint.

Reviewer 2 Report
Dear author,
your paper investigates the effect of selective retina therapy using
automatic real-time feedback-controlled dosimetry for CSC.
It is a well written paper on this topic that holds reliable information
for the reader.
Please find my remarks below:
-Introduction: Please add a paragraph regarding spontaneous recovery in CSC.
-Methods: A classification of your subjects based on RPE changes visualized using
fundus autofluorescence as described in Lee et al. 2016 (see below) should be added for comparison
to previously published articles using the same classification.
-Results: Adding three month results would allow a better comparison of the provided
data with the literature.
-Discussion: Please discuss that the shown data was collected in a single center study that
might hold some BIAS.
"Lee, W., Lee, JH. & Lee, B. Fundus autofluorescence imaging patterns
in central serous chorioretinopathy according to chronicity. Eye 30,
1336–1342 (2016). https://doi.org/10.1038/eye.2016.113"
Author Response
Reviewer 2
Dear author,
your paper investigates the effect of selective retina therapy using automatic real-time feedback-controlled dosimetry for CSC. It is a well written paper on this topic that holds reliable information for the reader.
Please find my remarks below:
-Introduction: Please add a paragraph regarding spontaneous recovery in CSC.
Response) We revised the sentence and added the reference. (Page 1, line 36)
Ref no.6) Klein,M.L., Van Buskirk, 1974 Experience with nontreatment of central serous chorioretinopathy Arch. Ophthalmol. 91, 247-250
-Methods: A classification of your subjects based on RPE changes visualized using fundus autofluorescence as described in Lee et al. 2016 (see below) should be added for comparison to previously published articles using the same classification.
"Lee, W., Lee, JH. & Lee, B. Fundus autofluorescence imaging patterns in central serous chorioretinopathy according to chronicity. Eye 30,1336–1342 (2016). https://doi.org/10.1038/eye.2016.113"
Response) Thank you for reviewer’s comments. We revised the method section (Page 4, line 156-161) and added the data of autofluorescence in Table 1 (Page 5, line 206-208). The reference paper was included.
Ref no.31) "Lee, W., Lee, JH. & Lee, B. Fundus autofluorescence imaging patterns in central serous chorioretinopathy according to chronicity. Eye 30,1336–1342 (2016).
-Results: Adding three month results would allow a better comparison of the provided data with the literature.
Response) We added the data of crossover group and retreatment group at 6-week and 12-week follow-up visits in Table 4. We revised results section. (Page 7, line 245-253)
-Discussion: Please discuss that the shown data was collected in a single center study that might hold some BIAS.
Response) We added the paragraph regarding the bias associated with a single center trial in discussion section. (Page 11, line 375-377) We added the reference.
Ref. no.38) Unverzagt, S.; Prondzinsky, R.; Peinemann, F. Single-center trials tend to provide larger treatment effects than multicenter trials: a systematic review. Journal of Clinical Epidemiology 2013, 66, 1271-1280.
